# Automated detection and prediction of suicidal behavior from clinical notes using deep learning

Brian E. Bunnell[1]*, Athanasios Tsalatsanis[2], Chaitanya Chaphalkar[2], Sara Robinson[1], Sierra Klein[1], Sarah Cool[1], Elizabeth Szwast[3], Paul M. Heider[3,4], Bethany J. Wolf[3], Jihad S. Obeid[3,4]

1 Department of Psychiatry and Behavioral Neurosciences, Morsani College of Medicine, University of South Florida, Tampa, Florida, United States of America, 2 Research Methodology and Biostatistics Core, Office of Research, Morsani College of Medicine, University of South Florida, Tampa, Florida, United States of America, 3 Biomedical Informatics Center, Medical University of South Carolina, Charleston, South Carolina, United States of America, 4 Department of Public Health Sciences, Medical University of South Carolina, Charleston, South Carolina, United States of America

* bbunnell@usf.edu

## Abstract

### Background

Deep learning approaches have tremendous potential to improve the predictive power of traditional suicide prediction models to detect and predict intentional self-harm (ISH). Existing research is limited by a general lack of consistent performance and replicability across sites. We aimed to validate a deep learning approach used in previous research to detect and predict ISH using clinical note text and evaluate its generalizability to other academic medical centers.

### Methods

We extracted clinical notes from electronic health records (EHRs) of 1,538 patients with International Classification of Diseases codes for ISH and 3,012 matched controls without ISH codes. We evaluated the performance of two traditional bag-of-words models (i.e., Naïve Bayes, Random Forest) and two convolutional neural network (CNN) models including randomly initialized (CNNr) and pre-trained Word-2Vec initialized (CNNw) weights to detect ISH within 24 hours of and *predict* ISH from clinical notes 1–6 months before the first ISH event.

### Results

In *detecting* concurrent ISH, both CNN models outperformed bag-of-words models with AUCs of.99 and F1 scores of 0.94. In *predicting* future ISH, the CNN models out-performed Naïve Bayes models with AUCs of 0.81–0.82 and F1 scores of 0.61−.64.

**Data availability statement:** The University of South Florida's Human Research Protections restricts data sharing to protect patient information and comply with local and national regulations due to sensitive patient information. Public sharing could compromise patient privacy. Researchers seeking access to the data can contact USF Human Research Protections (IRB) at RSCH-IRB@usf.edu referencing the study number STUDY002988.

**Funding:** This project was supported in part by the National Institute of Mental Health grant number R56 MH124744 and the National Center for Advancing Translational Sciences under Grant Number UL1 TR001450. The content is solely the responsibility of the authors and does not necessarily represent the official views of the National Institutes of Health. the funders had no role in study design, data collection and analysis, decision to publish, or preparation of the manuscript.

**Competing interests:** The authors have declared that no competing interests exist.

## Conclusions

We demonstrated that leveraging EHRs with a well-defined set of ISH ICD codes to train deep learning models to detect and predict ISH using clinical note text is feasible and replicable at more than one institution. Future work will examine this approach across multiple sites under less controlled settings using both structured and unstructured EHR data.

## Introduction

Suicide has consistently been among the top ten leading causes of death in the U.S. during the past decade, resulting in more than 480,000 deaths, with rising rates each year [1]. These suicidal deaths and related nonfatal suicide-related injuries are associated with significant economic burden in the U.S., with annual medical costs and associated loss in productivity reaching over $500 billion [2]. There are numerous risk factors for suicide, some of which include being male, American Indian or Alaska Native, non-Hispanic, and having mental illness (e.g., depression, anxiety, substance abuse), prior trauma, communication difficulties, decision-making impulsivity, and aggression [1]. Other important risk factors include prior suicide attempts or intentional self-harm (ISH) behaviors [3,4]. Although many risk factors have been identified, meta-analytic data suggest that they only provide a marginal improvement in diagnostic accuracy above chance and prospectively predict suicide attempts only 26% of the time and suicide deaths only 9% of the time [4,5].

Identifying individuals at risk for suicide is critical to suicide prevention [6]. However, current approaches to suicide risk assessment involve healthcare professionals administering clinical interviews and questionnaires, which can be inefficient, costly, and limited in their ability to predict future ISH and suicide deaths [5,7–11]. Analytical approaches, such as machine learning that make use of data from electronic health records (EHRs), have tremendous potential to improve the efficient identification and predictive accuracy of numerous risk factors without having to repeat clinical interviews and questionnaires across multiple treatment settings. For example, machine learning approaches analyzing structured EHR data [12–18] and natural language processing (NLP) of clinical note text (e.g., word pair frequencies, positive or negative valence words) [12,19–26] to predict suicide and ISH have shown promising, yet variable results, with area under the receiver operating characteristic curves (AUCs) ranging between 0.61 and 0.89.

Recent advances in computational approaches, such as deep learning, have the potential to significantly improve suicide/ISH prediction and consistency in predictive performance by harnessing contextual content, rather than simple word frequencies within clinical notes [27,28]. Deep learning encompasses computational models composed of multiple layers of artificial neural networks, or deep neural network (DNN) based models, such as convolutional neural networks (CNNs), that "learn" representations of data with multiple levels of abstraction [29]. Several studies have examined innovative deep learning approaches to detect suicidality. For example, studies

have successfully employed deep learning using text from social media posts to detect psychiatric stressors for suicide [30], predefined suicide risk categories [31], and users with signs of suicidal ideation [32,33]. Another novel study used deep learning and video-recorded interviews of suicidal patients to detect suicide severity from digitally quantified measurements of facial expressivity, head movement, and speech prevalence [34]. However, despite the innovative nature of these studies, they did not capitalize on the abundance of valuable and relevant information within EHRs, which has the potential to improve risk assessment.

Currently, few studies have examined the utility of deep learning approaches to identify suicide-related clinical records using clinical text from an EHR (e.g., for surveillance purposes) [27,35–37], only one of which examined its utility to predict future suicidal behavior [27]. Cusick and colleagues [35] used deep learning to detect current suicidal ideation from EHR clinical note text and found their CNN model to be superior to other machine learning methods (i.e., logistic classifier, support vector machines [SVM], Naive Bayes classifier), demonstrating an AUC of 0.962. Rozova et al. [36] used several machine learning approaches to detect instances of ISH based on text from emergency department triage notes while also implementing a long short-term memory (LSTM) network but found that the LSTM approach performed poorer than a gradient boosting model (i.e., AUCs of 0.801 *vs*. 0.832, respectively). Using clinical notes from the U.S. Veterans Affairs, Workman et al. [37] found that a trained zero-shot learning DNN model was able to effectively identify ICD-10-CM diagnostic codes related to suicide with an AUC of .946. Obeid and colleagues [27] compared the performance of two DNN approaches (i.e., CNN and LSTM) with several traditional bag-of-words models (e.g., naïve Bayes, decision tree, random forest) in identifying current ISH and predicting future ISH events from notes 1–6 months earlier. The CNN approach achieved the highest performance in detecting current ISH (i.e., AUC = 0.999, F1 score = 0.985) and predicting future ISH (i.e., AUC = 0.882; F1 score = 0.769). Despite the promising results in these machine learning and deep learning approaches, they are limited by a general lack of consistent performance and replicability of results [27,29–31,33–36,38]. Furthermore, although the report by Obeid et al. included an examination of overrepresented words in clinical notes of patients with ISH events, it did not include an interpretation of why some clinical notes were indicative of ISH.

Given the limited prior work evaluating deep learning approaches using EHR clinical note text for the detection and prediction of ISH, there is a critical need to conduct more studies that exhibit stronger predictive power and show consistent model performance across multiple data sets. Thus, the purpose of the current study was to validate the approach used previously by Obeid et al. [27], specifically, to replicate the results at another institution [39], and to evaluate its generalizability to other academic medical centers. Of note, this approach leverages medical records coded with a well-defined set of ISH International Classification of Diseases, Clinical Modification (ICD) codes to generate large amounts of labeled records to train machine learning models using supervised learning [40]. This relies on the quality and accuracy of the ICD coded records. Specifically, herein, we aimed to assess the accuracy of ISH ICD codes at another institution and assess their value as "silver standard" labels for training models on the following two tasks: 1) automated detection of suicide attempt and ISH events in clinical note text concurrent with documented codes for ISH (hereafter referred to as concurrent ISH), and 2) the prediction of future ISH-labeled encounters. We also aimed to examine the interpretability of some models by identifying key words in clinical notes that may be indicative of suicidal behavior. We envision that the proposed deep-learning approach will benefit current suicide risk assessment by providing additional valuable information from existing clinical documentation that is unavailable to a provider during the patient's assessment, as it may not be included in the patient's primary problem list.

## Methods

### Participants

Adult patients, ages 18–90 years old, with clinical notes in the Epic (Epic Systems Corporation) EHR system at the University of South Florida (USF) or its affiliate Tampa General Hospital (TGH) between August 2015 and August 2021 were eligible for inclusion in the study. Cases included all patients with ICD-10 codes for a suicide attempt (i.e., T14.91) or ISH

as defined in the Centers for Disease Control and Prevention's National Health Statistics Report (e.g., X71–X83) [40]. Controls included a cohort of randomly selected contemporary patients without ICD codes for a suicide attempt or ISH diagnosis in their records during the study period. We first pulled 5 times more controls than cases from the organization's EHR. The controls were then matched to cases by age, gender, race, ethnicity, and number of notes using the nearest neighbor approach, such that 2 controls per case were selected for the study. The index event for cases was defined as the first suicide attempt or ISH diagnosis, and the index event for controls included the most recent interaction with the EHR. Clinical notes up to 180 days before and including the index event day were extracted. The study was approved by the USF Institutional Review Board (IRB) under the STUDY002988 protocol. The IRB waived the requirement for informed consent. The data were accessed for research purposes on August 12, 2022. Some authors had access to information that could identify individual participants during or after data collection.

## Clinical note selection and preprocessing

**Note types.** While all types of clinical notes were extracted from the EHR, to reduce noise and excessive number of notes, we excluded types with small frequencies and emphasized types such as Emergency Department (ED), Progress, Plan of Care, ED Provider, Consults, Interdisciplinary Plan of Care, History and Physical, Case Management, and ED After Visit Summary (AVS) Snapshot (see S1 Table).

**Preprocessing of notes for concurrent detection of ISH.** Notes recorded within 24 hours (i.e., +/-24hrs) of the index event were used to (1) assess the reliability of ICD codes in capturing suicide attempts or ISH events by manual review of the notes and (2) detect suicide attempts or ISH using machine learning models. Following the procedures used by Obeid et al. (2020), individual notes longer than 800 words were truncated to 800 words (n = 2,793). Notes belonging to the same patient were sorted by timestamp with newer notes first and then concatenated. Concatenated notes longer than 8,000 words were truncated to 8,000 words (n = 38). The notes were stored in the Detection cohort dataset, in which, each record represented a patient and their concatenated note. Patients without notes within 24 hours of the index event were not included in the Detection cohort dataset.

**Preprocessing of notes for predicting ISH.** Notes recorded between 30 and 180 days before the index event were used for the prediction of suicide attempts or ISH events using machine learning models. Individual notes were truncated to 1500 words (n = 7,545), and concatenated notes were truncated to 10,000 words (n = 744). The rationale for using longer word cutoffs was to capture more information in the longitudinal record prior to the ISH visit than we did with the Detection cohort. The notes were stored in the Prediction dataset, in which each record represented a patient and their concatenated note. Patients without notes in the 180-to-30-day window were not included in the Prediction cohort dataset.

## ICD validation through manual review

A sample of 400 patient records from the Detection cohort dataset was manually reviewed to assess the reliability of ICD codes in capturing suicide attempt or ISH events reported in clinical notes. The sample included records from 200 cases and 200 controls. The notes were imported into REDCap [41] and reviewed by two medical students blinded to the related ICD code. The reviewers were instructed to read the concatenated notes and label them as case (ISH) or control (no ISH). Suicidal ideation alone was not labeled as a case. Each reviewer was assigned 250 note strings with 100 overlapping to allow for estimation of interrater reliability. Any conflicting ratings were resolved by a licensed Clinical Psychologist (i.e., the first author). Reviewer labels were considered the gold standard and were compared with the suicide attempt or ISH ICD codes.

## Text processing and word embeddings

We tested three machine learning models: a deep learning model with word embeddings (WEs) and two traditional bag-of-words models. For the deep learning model, we performed the following functions in the concatenated notes of the Detection and Prediction cohort datasets: lower casing; sentence segmentation; removal of punctuation and numbers;

and tokenization. To create word embeddings, token sequences were pre-padded with zeros to match the length of the longest string in the training set. WE had 200 dimensions per word. WE weights were initialized a) randomly and b) with a pretrained Word2Vec (W2V) [42] model, built on 550,000 notes from the EHR using 200 dimensions per word, a skip window size of 5 words in each direction, and negative sampling of 5. For bag-of-words models, text processing included lower casing; removal of punctuation, stop word, and numbers; word stemming, and tokenization.

**Bag-of-words models**

The models we tested were Naïve Bayes (NB) [43] and a Random Forest (RF) [44] with 1000 trees sampling 700 variables per split.

**Deep learning models**

The CNN architecture consisted of several layers (Fig 1). We used an architecture that has been previously developed and used for text classification [28,45]. The input layer had a dimension of 10,000 tokens. The next layer was a WE with a drop rate of 0.2. Next, there was a convolutional layer with multiple filter sizes (3, 4, and 5) in parallel, with 200 nodes in each, ReLU activation, a stride of one, and a global max-pooling followed by a merger tensor. Then, a fully connected 200-node hidden layer with ReLU activation and a drop rate of 0.2. Lastly, an output layer with a single binary node with a sigmoid activation function. The CNN training parameters were as follows. The optimizer used: adaptive moment estimation gradient descent algorithm (ADAM); number of epochs: 37; batch size: 32; learning rate:.0004; validation spit: 0.1; and early stopping based on validation loss function patience of 5. We tested the architecture with randomly initialized weights and with W2V-initialized weights.

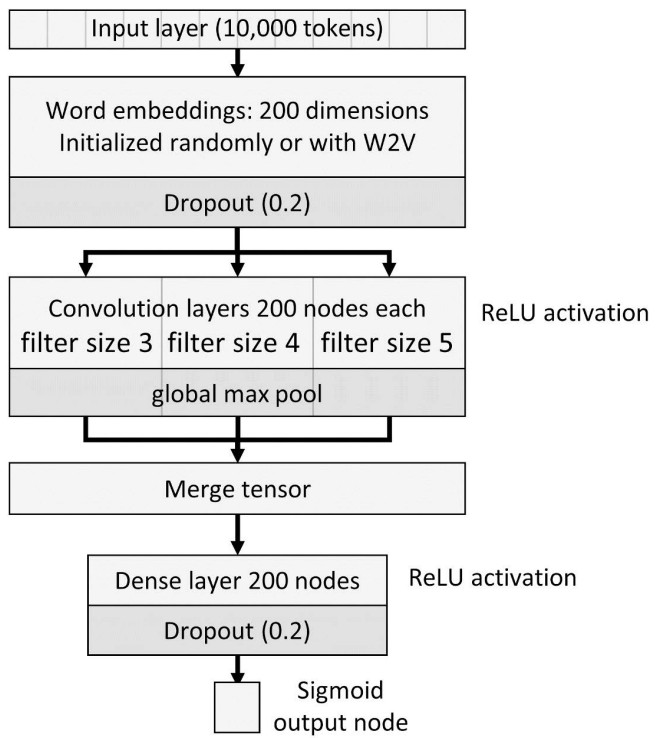

**Fig 1. The CNN architecture.** ReLU: rectified linear units activation function; W2V: word2vec embeddings.

## Software packages

The R Statistical Software ver. 4.0.2 was used in all analyses [46]. To match controls and cases, we used the MatchIt package [47]. The Quanteda package [48] was used for text processing and for the NB model. The Keras package [49] was used to initialize WEs. W2V was used to initialize WE pre-trained weights [42]. The CNN was constructed using TensorFlow [50]. The RF was constructed using the ranger package [51]. Metrics were calculated using the Caret [52] and pROC [53] packages.

## Evaluation and metrics

For the ICD manual review, we calculated and reported false positive and false negative counts, interrater reliability (Cohen's Kappa), accuracy, precision, and recall. In preparation for the machine learning models, both Detection and Prediction cohort datasets were split into a training and cross-validation set including cases and controls with an index event occurring between 2015 and 2019, and a testing set including cases and controls with an index event occurring between 2020 and 2021. For all classification models, we calculated the AUC, accuracy, precision, recall, and F1 score. The evaluation metrics of the CNN models were based on the median values of 50 runs. The DeLong test was used to determine possible significant differences between performances of models [54]. We also reported variable (stemmed word) importance calculated by the RF model. Finally, we report the performance of the CNN models against the gold standard as defined by the manual review of 400 records.

## Results

### Patient population

Data from 11,298 patients (i.e., 1,883 cases and 9,415 controls) were extracted from the EHR, as per S1 Fig. A total of 6,814 patients (i.e., 1,538 cases and 5,276 controls) had notes recorded 24 hours within the index event (i.e., the Detection cohort). Of these, 1,593 cases were matched with 3,012 controls (N = 4,605). A total of 3,474 patients (i.e., 1,158 cases and 2,316 matched controls) with an index event between 2015 and 2019 were selected for the *training and cross-validation* of the CNN used for detecting suicide attempts or ISH. Lastly, 1,076 patients (i.e., 380 cases and 696 matched controls) with an index event between 2021 and 2021 were selected to *test* the CNN.

A total of 7,925 patients (i.e., 593 cases and 7,332 controls) had notes recorded between 30 and 180 days prior to the index event. Of these, 593 cases were matched with 1,186 controls (N = 1,779). A total of 487 cases with an index event between 2015 and 2019 and 974 matched controls (N = 1,461) were selected for the *training and cross-validation* of the prediction model. The 106 cases with an index event between 2020 and 2021 and 212 matched controls (N = 318) were selected for the prediction model testing (see S1 Fig).

S2 Table shows the demographic characteristics of cases and controls from the Detection and Prediction cohorts used to develop and test the CNN models. The Detection cohort was largely white (i.e., 64% of cases and 64% of controls) and non-Hispanic/Latino (i.e., 81% of cases and 80% of controls). There was a statistically significant difference in age (Mann-Whitney U = 1,782,666, n1 = 1,538, n2 = 3,012, $p < 0.001$) and sex ($x^2$ (1, N = 4,550) = 126.270, $p < 0.001$) between cases and controls. The case group was younger on average and had more males (55%) than the control group (48%). There were no significant differences in race ($x^2$ (2, N = 4,550) = 0.1632, $p = .921$), or ethnic background ($x^2$ (2, N = 4550) = 4.4055, $p = .110$) between cases and controls.

The Prediction cohort was largely female (i.e., 57% of cases and 60% of controls), white (i.e., 67% of cases and 69% of controls), and non-Hispanic/Latino (i.e., 84% of cases and 85% of controls). There were no significant differences in age (Mann-Whitney U = 349,408, n1 = 593, n2 = 1,186, $p = 0.826$), sex ($x^2$ (1, N = 1,779) = 1.596, $p = .207$), race ($x^2$ (2, N = 1,779) = 0.126, $p = .937$), or ethnicity ($x^2$ (2, N = 1,779) = 0.079, $p = .961$) between cases and controls.

## Clinical note characteristics

S1 Table lists the type of notes used in the analyses and frequency per study group. In the Detection cohort, the most frequent note type for cases was Emergency Department (ED) Notes, while the most frequent note type for controls was Progress Notes. On average, there were 6 times more notes per person for cases than controls (11.6 vs. 1.98 notes per person) in the detection cohort. In the Prediction cohort, the most frequent note type was Progress Notes for cases and controls. On average, controls had approximately five times more notes than cases (i.e., 163.5 *vs.* 36.4 notes per person) in the prediction cohort.

## ICD validation through manual review

The interrater reliability for the manual review of clinical notes recorded within 24 hours after the index event was 0.986. In the sample of 400 patients (200 cases and 200 controls), 39 were diagnosed with ICD codes for suicide attempts or ISH when it was not documented in their clinical notes (i.e., false positives), and 2 patients were not assigned ICD codes for suicide attempts or ISH when their clinical notes indicated they should have (i.e., false negatives). Overall, the accuracy of the ICD codes against the manual review was 0.90, Precision was 0.80, and Recall was 0.99.

## Detection of concurrent ISH

The performance of the machine learning models in detecting suicide or ISH events from clinical notes recorded within 24 hours of the index event is presented in Table 1. Fig 2 displays the receiver operating characteristic (ROC) curves for the different models. All models performed well with the ISH detection task with AUCs over 0.94. In general, CNN models outperformed the bag-of-word models (i.e., NB and RF), with $p$-value $< .05$ using the DeLong test. When rounded to two digits after the decimal, both CNN models (whether WEs were randomly initialized or initialized with W2V) had very similar performance with no significant differences ($p$-value approaching 1). Both showed an AUC of 0.99 and an F1 score of 0.94. The NB model (Table 1 NB) demonstrated an AUC of 0.95 and an F1 score of 0.86. Lastly, for RF (Table 1 RF) AUC was 0.98 and the F1 score was 0.92. Accuracy, Precision (or Positive Predictive Value), Recall (or Sensitivity), Specificity, 95% AUC confidence intervals, and significance of the difference in AUC for all models are all available in Table 1.

**Table 1. Performance of the ML models in Detecting and Predicting Suicide/ISH. The highest metrics are bolded.**

**Models Detecting Concurrent ISH Events**

| Model | AUC (95% CI) | Accuracy | Precision | Recall | F1-score | Specificity | AUC Significantly> |
|---|---|---|---|---|---|---|---|
| NB | 0.947 (0.933-0.961) | 0.894 | 0.801 | 0.932 | 0.861 | 0.874 | |
| RF | 0.981 (0.973-0.989) | 0.946 | 0.924 | 0.924 | 0.924 | 0.958 | NB |
| CNNr | 0.986 (0.980-0.993) | 0.956 | 0.926 | 0.953 | 0.939 | 0.958 | NB, RF |
| CNNw | **0.987 (0.979-0.994)** | **0.957** | **0.928** | **0.953** | **0.940** | **0.960** | **NB, RF** |

Models **Predicting** Future ISH Events

| Model | AUC (95% CI) | Accuracy | Precision | Recall | F1-score | Specificity | AUC Significantly> |
|---|---|---|---|---|---|---|---|
| NB | 0.757 (0.701-0.813) | 0.745 | 0.634 | 0.557 | 0.593 | 0.840 | |
| RF | 0.790 (0.734-0.846) | 0.789 | **0.800** | 0.491 | 0.608 | **0.939** | |
| CNNr | **0.817 (0.768-0.866)** | **0.792** | 0.756 | **0.557** | **0.641** | 0.910 | NB |
| CNNw | 0.812 (0.763-0.861) | 0.774 | 0.718 | 0.528 | 0.609 | 0.896 | NB |

Detecting **Concurrent** ISH Events based on the Gold Standard (manual review of 400 notes)

| Model | AUC (95% CI) | Accuracy | Precision | Recall | F1-score | Specificity | |
|---|---|---|---|---|---|---|---|
| CNNr | 0.950 (0.915-0.986) | 0.902 | 0.93 | 0.866 | 0.897 | 0.936 | |
| CNNw | 0.956 (0.923-0.989) | 0.912 | 0.942 | 0.877 | 0.908 | 0.947 | |

NB: Naïve Bayes model; RF: Random Forest model; CNNr: Convolutional neural network model with randomly initialized word embeddings; CNNw: Convolutional neural network model with W2V initialized embeddings; significant differences in AUC were determined using DeLong test.

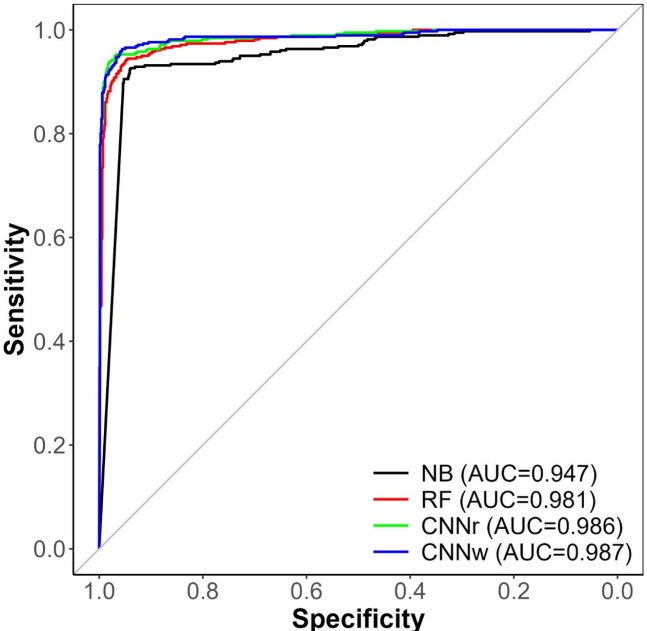

**Fig 2. ROC of models for suicide/ISH Detection.** CNN 1: Convolutional neural network model with randomly initialized word embeddings; CNN 2: Convolutional neural network model with W2V initialized embeddings; NB: Naïve Bayes model; RF: Random Forest model.

Fig 3 depicts the first 15 important stemmed words determined using variable importance analysis of the RF model. The first two most important stemmed words for classification in the list were "ed" and "suicid", followed by three words related to Florida's Baker Act (legislation requiring a means of providing individuals with emergency services and temporary detention for up to 72 hours for mental health examination): "act", "ba", and "baker".

### Prediction of future ISH events

The performance of the machine learning models in the prediction of suicidal behavior or ISH events from clinical notes recorded 30–180 days before the index event is presented in Table 1. Fig 4 displays the ROCs for all models. Here again, the CNN models outperformed both the NB and RF models. The CNN model with randomly initialized WEs (CNNr) demonstrated an AUC of 0.82 and an F1 score of 0.64. The CNN model with WEs initialized from the pre-trained W2V (CNNw) demonstrated an AUC of 0.81 and an F1 score of 0.61. The NB model (NB) demonstrated an AUC of 0.76 and an F1 score of 0.59. Lastly, for the RF model (RF), the AUC was 0.79 and the F1 score was 0.61. Using the DeLong test, both CNN AUCs were significantly higher than those of NB ($p < 0.05$). However, there were no significant differences between either of the CNNs or the RF, and no significant differences between the RF and NB. Unlike the CNNw model for the detection task, which was numerically higher across all scores, the CNNr model for the prediction task was higher for all metrics except Precision and Specificity, for which the RF model was highest.

S2 Fig shows the first 15 important words measured by the RF. The first three most important stemmed words for classification in the list were "mdm" (i.e., Medical Decision Making), "ed", and "suicid".

### Detection of concurrent ISH based on gold standard labels

The performance of the CNN models on detecting concurrent ISH based on the gold standard labels is presented in Table 1. The CNN model with randomly initialized WEs (CNNr) demonstrated an AUC of 0.95 and an F1 score of 0.897. The

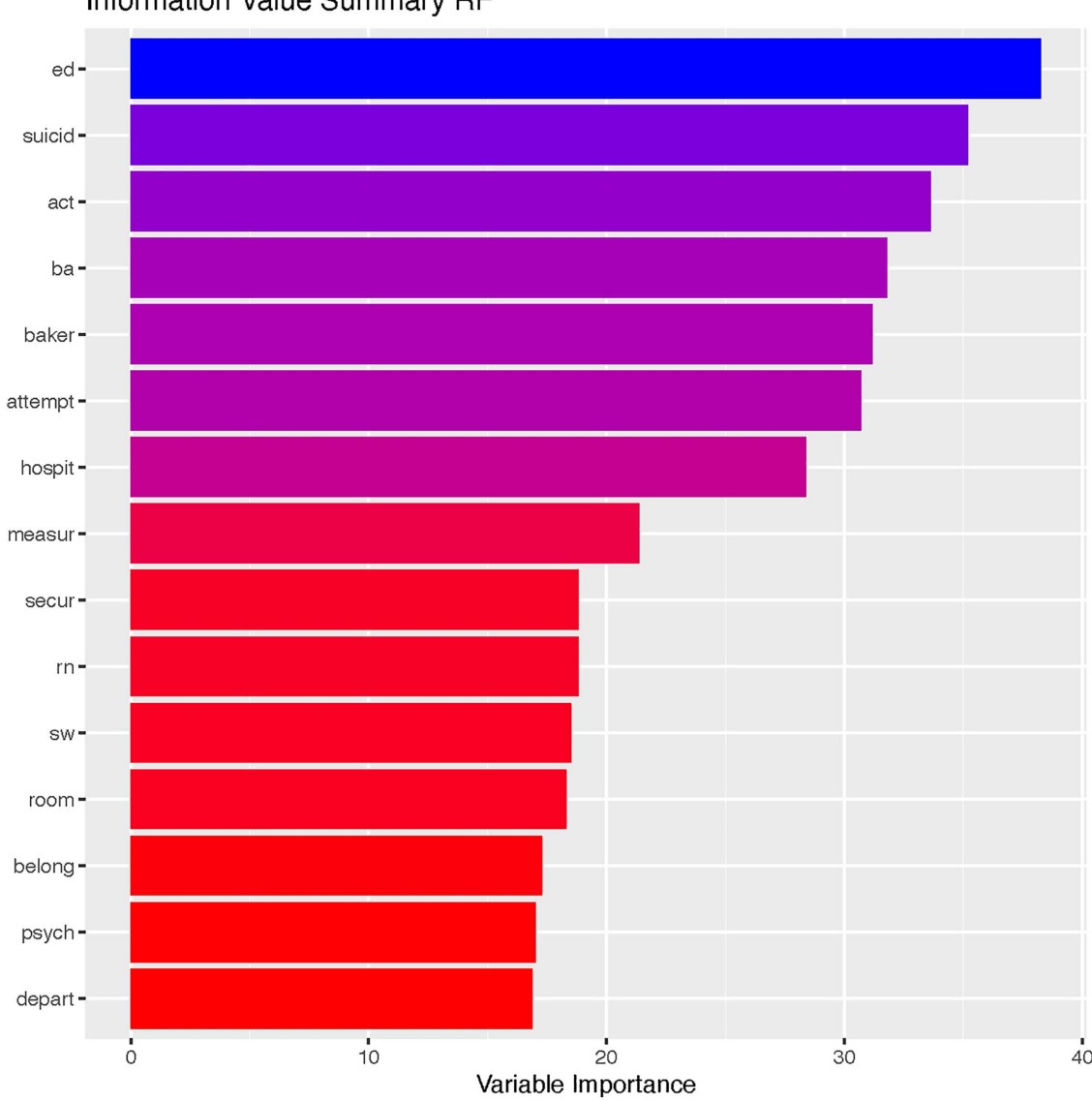

**Fig 3. Variable importance as measured by the RF for suicide/ISH Detection.** "Act, "ba", and "baker" likely relate to Florida's Baker Act, legislation surrounding mental health crisis. "Rn" and "sw" are common abbreviations for providers (i.e., registered nurse and social worker).

CNN model with WEs initialized from the pre-trained W2V (CNNw) demonstrated an AUC of 0.956 and an F1 score of 0.908.

## Discussion

Computational approaches such as deep learning have strong potential to improve ISH detection and prediction as well as consistency in predictive performance, but studies to date have been limited by a general lack of consistent performance and replication of the process [27,29–31,33–36,38]. The benefit of such deep learning models is that they can add value to the predictive power of traditional suicide predictive models. In this study, we demonstrated that leveraging medical records with a well-defined set of ISH ICD codes with a reasonably high accuracy (0.90 in our study) as "silver

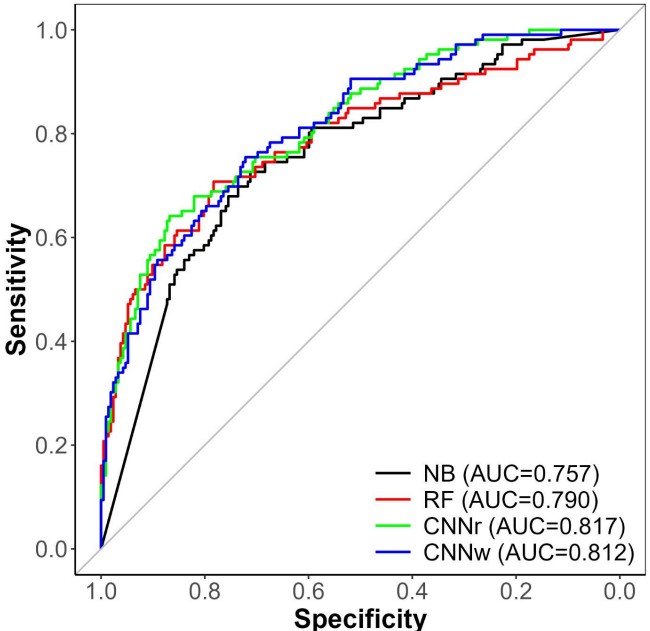

**Fig 4. ROC of models for the prediction of suicide/ISH.** CNN 1: Convolutional neural network model with randomly initialized word embeddings; CNN 2: Convolutional neural network model with W2V initialized embeddings; NB: Naïve Bayes model; RF: Random Forest model.

standard" labels to train deep learning models to detect and predict ISH using clinical note text is feasible and replicable at more than one institution. The results of this study are compatible with our previous work at another institution, where we compared different deep learning models using a similar temporal validation approach (i.e., training on data from 2012–2017 and testing on data from 2018–2019) [27]. As in this work, the models had a near-ceiling performance on the phenotyping task in that the automated identification of concurrent ISH events in clinical notes *vs.* a comparable AUC of 0.82 for the predictive task, suggesting that this approach could be replicated at other academic medical centers for both surveillance purposes and the identification of patients at risk of future suicidal behavior. Further upon closer examination of the false positives and false negatives by ICD as compared to manual review, we noted that the CNN classification was correct in 5 of the 39 ICD false positives and 1 of the 2 ICD false negatives, which suggests that the deep learning model could provide more precise retrieval of cases, e.g., for surveillance tasks, beyond simple ICD queries. It is worth noting that although the F1 scores seem low (0.64), the predictive model's performance (especially as highlighted with the AUC) is fairly competitive with what has been previously reported in the literature using different approaches with EHR data, including structured data; however, consideration should be given to the different type of predictive modeling previously used as well as different settings and feature sets. Nonetheless, one can calibrate the threshold to improve recall/sensitivity vs. the specificity depending on the setting. For example, to improve the recall, we could reduce the threshold to pick up more cases for referrals. The high AUC suggests that we are more likely to find a good balance for a given use case or environment in which such models could be used. In critical applications like suicide prediction, maximizing recall is critical. By lowering the probability threshold, we can increase the likelihood of identifying most patients at risk, even if it means having more false positives. We should note, however, that it is also crucial to continuously evaluate and refine the models, incorporating feedback from healthcare professionals and data from real-world outcomes to improve its accuracy and effectiveness over time.

The variable importance analysis of the RF model in both the detection of concurrent events and the predictive models provided some insight into words in clinical notes that are indicative of ISH. Detected important variables included word

stems that could potentially be generalizable to other sites, such as "suicid", "attempt" and "ed", as well as words that are relevant only in the local context, such as "ba" as in the Baker Act, which are specific to Florida. These words could be highlighted during the implementation of such models to draw the attention of clinicians in busy settings (e.g., primary care settings) and instill confidence and trust by clinicians in the results of risk prediction, with the increased likelihood of referral to mental health services for appropriate management [5,55–57].

Along these lines, a limitation of the bag-of-words models is that they rely on instances of single words (unigrams). Therefore, in a situation where a clinician documents the absence of an event (i.e., not suicidal), the model may misclassify the note. However, these models still evaluate the whole note by assessing the frequencies of other words relevant to ISH that increase the probability that a patient is, in fact, an ISH patient. Moreover, with deep learning models (i.e., CNNs), the sequence of the words (or tokens) is preserved, which has a greater potential of picking up such negations.

## Clinical implications

Current approaches to suicide risk assessment can be inefficient, costly, and limited in their ability to predict future ISH and suicide deaths [4,7,8,10,11]. As such, machine and deep learning methods that can improve the efficient and accurate identification of individuals who engage in or are likely to engage in ISH behavior have tremendous potential clinical benefits. For example, EHR systems that integrate these advanced analytical approaches can alert providers across a patient's care team to possible risk without relying on the repeated administration of clinical interviews and questionnaires, and encourage appropriate referral, intervention, and follow-up over time. This would be especially useful in cases where ISH risk is documented in prior clinical notes but is not added to a patient's primary problem list, and therefore not easily viewable by providers who were not the original documenter. Similarly, these approaches can be used to analyze data from multiple clinical encounters, accounting for risk factors documented by some providers and not others, resulting in a more complete synthesis of a patient's risk. This would be useful for informing patient risk assessment–including quantifying risk levels–as well as the observation of those risk levels over time. Therein also lies the potential for data-driven clinical decision support tools that inform the development of individualized treatment plans. Lastly, from a clinical and research perspective, they provide an opportunity to improve our understanding of how ISH and suicidal behavior risk can change over time to inform prevention and intervention initiatives on an individual- and systems-level.

## Limitations

This study has limitations that provide opportunities for future research. First, although we were able to show similar results to the original work conducted by Obeid and colleagues [27] at an additional medical center using the same methodology, it is necessary to show replicability across multiple sites. Ideally, we would also want to perform external validation of a trained model (i.e., train at one institution and test the trained model at another site), which was not feasible in this study due to funding limitations and regulatory restrictions. This presents a valuable opportunity to use smaller datasets to train and fine-tune the models. Moreover, although we demonstrated that we can clearly identify ISH, this does not specify an intention to die, so future work will need to examine this approach while incorporating data on fatalities resulting from suicide. Although the variable importance analysis sheds light on important key words in text, this work would benefit from further exploration of interpretable modeling approaches, especially regarding the deep learning models, which are currently thought of as "black boxes". However, previous work using hierarchical attention networks [57] seems promising for highlighting text that contributes to the outcome of the classifier. Another limitation is the relatively small sample size that we worked with for deep learning modeling. Having a larger sample would allow further examination in predictive time windows, for example, looking at clinical notes 6 months prior to the index visit, vs. 4 months, etc. We chose to use a 5-month-long time window between 6 months and 1 month prior to the index visit. Reducing the predictive time window to examine the temporal impact prior to the index event would require more stringent inclusion criteria and thus reduce the number of patients needed for each prediction. Further, our case-control matching artificially reduces the non-ISH to ISH

ratio, which may not reflect real-world scenarios and may artificially improve the reported accuracy and AUC. Lastly, our approach used text from clinical notes found in the EHR but did not include the integration of structured data which are also available in the EHR and have strong potential to strengthen the predictive power of the models.

### Future directions

There are several directions for this work that our team plans to pursue in the near future. The first is to show replicability and external validation across multiple sites using novel transfer learning methods. We also plan to examine more advanced deep learning models based on transformer architectures, including newer embedding models and large language models, which require a more advanced compute infrastructure than was used with the models described herein. Furthermore, we intend to compare the text-based predictors with traditional models using structured EHR data (e.g., history of previous mental health codes, medications, and other variables; [18]) as well as datasets with patients at high risk for ISH (e.g., with a history of depression). Lastly, as part of these efforts, we will include data on suicide deaths to examine the predictive power of our approach in the context of this outcome while also integrating explanatory modeling approaches.

### Conclusions

In this study, we demonstrated that leveraging medical records with a well-defined set of ISH ICD codes to train deep learning models to detect and predict ISH using clinical note text is feasible and generalizable given replicability at another institution in a different region of the U.S. Our results were similar to our prior work where we compared different deep learning models using a similar temporal validation approach. These approaches can improve the efficient and accurate identification and prediction of individuals who engage in ISH behavior and have enormous potential benefits for clinical practice. However, further research is needed to evaluate this approach across multiple sites and in less controlled, real-world settings, incorporating both structured and unstructured EHR data, as well as suicide fatality outcomes—despite the inherent challenges of modeling highly imbalanced outcome variables.

### Supporting information

**S1 Fig. Flowchart of study population.**
(TIF)

**S2 Fig. Variable importance as measured by the RF for the prediction of suicide/ISH.**
(TIF)

**S1 Table. Clinical note types and frequency per group.**
(DOCX)

**S2 Table. Participant demographics.**
(DOCX)

### Author contributions

**Conceptualization:** Brian E. Bunnell, Athanasios Tsalatsanis, Paul M. Heider, Jihad S. Obeid.

**Data curation:** Brian E. Bunnell, Athanasios Tsalatsanis, Chaitanya Chaphalkar, Sara Robinson, Sierra Klein, Sarah Cool, Elizabeth Szwast, Paul M. Heider, Bethany J. Wolf, Jihad S. Obeid.

**Formal analysis:** Brian E. Bunnell, Athanasios Tsalatsanis, Paul M. Heider, Jihad S. Obeid.

**Funding acquisition:** Brian E. Bunnell, Jihad S. Obeid.

**Investigation:** Brian E. Bunnell, Athanasios Tsalatsanis, Paul M. Heider, Jihad S. Obeid.

**Methodology:** Brian E. Bunnell, Athanasios Tsalatsanis, Chaitanya Chaphalkar, Sara Robinson, Sierra Klein, Sarah Cool, Elizabeth Szwast, Paul M. Heider, Bethany J. Wolf, Jihad S. Obeid.

**Project administration:** Brian E. Bunnell, Elizabeth Szwast, Jihad S. Obeid.

**Resources:** Athanasios Tsalatsanis, Jihad S. Obeid.

**Software:** Athanasios Tsalatsanis, Jihad S. Obeid.

**Supervision:** Brian E. Bunnell, Athanasios Tsalatsanis, Jihad S. Obeid.

**Validation:** Brian E. Bunnell, Athanasios Tsalatsanis, Paul M. Heider, Jihad S. Obeid.

**Visualization:** Brian E. Bunnell, Athanasios Tsalatsanis, Jihad S. Obeid.

**Writing – original draft:** Brian E. Bunnell, Athanasios Tsalatsanis, Paul M. Heider, Jihad S. Obeid.

**Writing – review & editing:** Brian E. Bunnell, Athanasios Tsalatsanis, Chaitanya Chaphalkar, Sara Robinson, Sierra Klein, Sarah Cool, Elizabeth Szwast, Paul M. Heider, Bethany J. Wolf, Jihad S. Obeid.

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
