## [Decision Letter · Decision Letter 0]

11 Oct 2024

PONE-D-24-31813Automated detection and prediction of suicidal behavior from clinical notes using deep learningPLOS ONE

Dear Dr. Tsalatsanis,

Thank you for submitting your manuscript to PLOS ONE. After careful consideration, we feel that it has merit but does not fully meet PLOS ONE’s publication criteria as it currently stands. Therefore, we invite you to submit a revised version of the manuscript that addresses the points raised during the review process.

We look forward to receiving your revised manuscript.

Kind regards,

Braja Gopal Patra, Ph.D.

Academic Editor

PLOS ONE

Journal Requirements:

1. When submitting your revision, we need you to address these additional requirements. Please ensure that your manuscript meets PLOS ONE's style requirements, including those for file naming. The PLOS ONE style templates can be found at https://journals.plos.org/plosone/s/file?id=wjVg/PLOSOne_formatting_sample_main_body.pdf and https://journals.plos.org/plosone/s/file?id=ba62/PLOSOne_formatting_sample_title_authors_affiliations.pdf 2. We note that the grant information you provided in the ‘Funding Information’ and ‘Financial Disclosure’ sections do not match.  When you resubmit, please ensure that you provide the correct grant numbers for the awards you received for your study in the ‘Funding Information’ section. 3. Thank you for stating the following financial disclosure: "This project was supported in part by the National Institute of Mental Health grant number R56 MH124744 and the National Center for Advancing Translational Sciences under Grant Number UL1 TR001450. The content is solely the responsibility of the authors and does not necessarily represent the official views of the National Institutes of Health." Please state what role the funders took in the study.  If the funders had no role, please state: "The funders had no role in study design, data collection and analysis, decision to publish, or preparation of the manuscript." If this statement is not correct you must amend it as needed. Please include this amended Role of Funder statement in your cover letter; we will change the online submission form on your behalf. 4. We note that you have indicated that there are restrictions to data sharing for this study. For studies involving human research participant data or other sensitive data, we encourage authors to share de-identified or anonymized data. However, when data cannot be publicly shared for ethical reasons, we allow authors to make their data sets available upon request. For information on unacceptable data access restrictions, please see http://journals.plos.org/plosone/s/data-availability#loc-unacceptable-data-access-restrictions.  Before we proceed with your manuscript, please address the following prompts: a) If there are ethical or legal restrictions on sharing a de-identified data set, please explain them in detail (e.g., data contain potentially identifying or sensitive patient information, data are owned by a third-party organization, etc.) and who has imposed them (e.g., a Research Ethics Committee or Institutional Review Board, etc.). Please also provide contact information for a data access committee, ethics committee, or other institutional body to which data requests may be sent. b) If there are no restrictions, please upload the minimal anonymized data set necessary to replicate your study findings to a stable, public repository and provide us with the relevant URLs, DOIs, or accession numbers. Please see http://www.bmj.com/content/340/bmj.c181.long for guidelines on how to de-identify and prepare clinical data for publication. For a list of recommended repositories, please see https://journals.plos.org/plosone/s/recommended-repositories. You also have the option of uploading the data as Supporting Information files, but we would recommend depositing data directly to a data repository if possible. Please update your Data Availability statement in the submission form accordingly. 5. Please include captions for your Supporting Information files at the end of your manuscript, and update any in-text citations to match accordingly. Please see our Supporting Information guidelines for more information: http://journals.plos.org/plosone/s/supporting-information.

Reviewers' comments:

Reviewer's Responses to Questions

**Comments to the Author**

1. Is the manuscript technically sound, and do the data support the conclusions?

Reviewer #1: Partly

Reviewer #2: Partly

Reviewer #3: Partly

2. Has the statistical analysis been performed appropriately and rigorously? 

Reviewer #1: Yes

Reviewer #2: Yes

Reviewer #3: I Don't Know

3. Have the authors made all data underlying the findings in their manuscript fully available?

Reviewer #1: No

Reviewer #2: Yes

Reviewer #3: Yes

4. Is the manuscript presented in an intelligible fashion and written in standard English?

Reviewer #1: Yes

Reviewer #2: Yes

Reviewer #3: Yes

5. Review Comments to the Author

Reviewer #1: 1. The authors need to more explicitly articulate why and how they demonstrate the generalizability of the methodology they apply.

I appreciate that the authors' objective is to reproduce the methodology of Obeid et al (2020), but it is not clear why it would be a challenge to reproduce the methodology given the existence of ICD-coded data and access to notes. Why is it interesting to investigate the reproduction of this method? What can we learn from this attempt? The authors do not explain this point or truly "validate" the reproduction of the method, as they claim is the core objective of the paper (p. 5). Is it enough to follow the same methodology and present results of a model built from that methodology to "evaluate [the methodology's] generalizability to other academic medical centers? Given that -- as the authors state -- the method hinges on the availability and quality of ICD coded records, that seems to bve the only point. Or is it that a certain expected level of performance of the model must be achieved to claim that the methodology is generalizable? It simply isn't clear what the criteria for "validating" the generalizability of the methodology are. How can we claim success on this goal?

2. The real-world applicability of the method is still not adequately addressed/discussed given artificial selection criteria when constructing the cohorts.

More importantly, the significance of the original methodology and the new application of that methodology in this new setting is unclear. As the authors themselves point out, it is more important to examine how well a trained model itself generalizes -- both to more real-world settings, and to other hospital sites. This is not tackled by the authors of this work. As I will mention below, the datasets used to train and test the authors' model do not appear to be fully realistic. The test data in particular would be much more interesting if it reflected the full complexity of the real-world data environment, including the low prevalence of ISH presentations overall.

I grant that the performance of the models is well beyond random chance, but overall it seems that there is significant uncertainty as to how well this model would work prospectively in the natural data environment, given the steps taken to artificially ensure a minimum number of Cases in the datasets (at least 1/3), while also not controlling for Controls that are the most similar to the Cases. Controls are "matched" only in terms of a superficial note profile, plus to control numbers, not to identify the most likely potential confusers for the model.

When constructing the cohort datasets, did you ensure that the same patients did not appear in both training and test data sets? i.e. not simply splitting at the event level, but rather at the patient level. This is important to control for data leakage.

The data also appears to include "ED After Visit Summary (AVS)" reports, which I suppose are something like a Discharge Summary. Given that this is very likely to have a complete review/summary at the end or after of an ED episode, it seems that this type of report should be excluded as it will not be relevant to detecting a patient with ISH while they are in care.

Why was a 1:5 ratio adopted for Cases:Controls? To what extent is this representative of the overall prevalence of ISH in the data? Then, why was a 1:2 ratio assumed for "matched controls" in the final training/testing cohorts? This results in training/test cohorts that are very far out of balance from the a priori probability of ISH in the overall dataset.

I assume that "Controls" are patients that have no (0) suicide attempts or ISH ICD Codes at all in the data, although this is not fully specified. Then you filter out Controls that meets the notes requirements.

- What is the "index event" that is considered for a Control?

- Significant differences in the types of clinical texts are reported for cases vs controls in both the Detection and Prediction cohorts. This arguably means that distinguishing between cases and controls is easier; it is not only the content of the notes that varies, but the a priori nature of the information they reflect. These differences also imply substantive differences in the nature of problems and patterns of care that the cases and controls may be attended for. There would presumably be much more extra-cases variability, than intra-cases variability, which again makes distinguishing between them potentially very easy.

- What is the reason for Cases having so many more notes than Controls (11.6 vs 1.98) -- is this because there are many (short) ED notes cf. fewer (long) Progress Notes for Controls?

- Given the predominance of ED notes for Cases cf Controls, why was a "matched control" not one that had a similar notes profile to a Case, in terms of type of notes, number of notes, length?

3. Clarify manual review process.

It is not fully clear what data the manual review of clinical notes includes -- does this mean up to 8000 words of text, as is prepared for the model? Or full chart review?

4. Gold standard needed for testing.

It is one thing to *train* supervised learning models with "silver standard" data. It is another to *test* the models on silver standard data. What did you do to ensure the reliability of the final results on the held-out test data?

Reviewer #2: In this manuscript the authors posit that deep learning approaches have great potential to significantly improve traditional suicide prediction models to detect and predict intentional self-harm and report that existing research is limited by a general lack of “consistent performance” and “external reproducibility”. The work in the paper is aimed to “validate a deep learning approach” used in previous research [reference 27] to detect and predict ISH using clinical note text and “evaluate its generalizability to other academic medical centers”.

The manuscript is well written and properly structured without significant obstacles to understandability in terms of language and format. The methodology is sound and well explained though the methodology is not the novel contribution of the manuscript and has been published and implemented in [27]. Therefore, as I understand it, the main contribution of this paper is applying this modeling process(methodology) to another data set also validated by the statement from the manuscript “Thus, the purpose of the current study was to validate the approach used previously by Obeid et al. (2020) and to evaluate its generalizability to other academic medical centers.” (page 5 of manuscript).

In the limitations section the authors indicate “This study has limitations that provide opportunities for future research. First, although we were able to reproduce the original work conducted by Obeid and colleagues (2020) at an additional medical center, it is necessary to show reproducibility across multiple sites. Ideally, we would also want to perform external validation of a trained model (i.e., train at one institution and test the trained model at another site).

I believe the following revisions will improve the manuscript:

1. The authors should go through the manuscript and clarify the distinction between model and process(methodology) validation and specify that the contribution of the paper is to process validation and not model validation.

2. The authors indicate that lack of consistent performance is an important limitation of existing research. While at least as far as process validation is concerned they are able produce consistent results, the authors are silent on what could be the potential cause of such variability. Is it the data used, the methodologies, a combination of the two? OR is it only normal that when such a wide net including structured EHR vs clinical note text with sub-populations such as those with eating-disorders, first time suicide attempts, etc. is considered, such variability is inevitable. I believe the authors possess the experience to address, or at a minimum discuss, the reason for this gap in existing research.

3. The audience would be greatly interested in understanding why a model validation (along with process validation) was not reported in the manuscript. Did the authors face difficulties or simply have not attempted it (to me it does not seem like a difficult extension)? As the authors themselves indicate, this is also very important, and the readers would appreciate their experience and perspective.

4. The cited literature is until 2022 and should be updated as we are approaching 2025.

5. The term “concurrent ISH” is used several times in the manuscript. The reader is not clear on the meaning of the word “concurrent” here.

6. On page 11, the section on “Clinical Note Types and Frequencies” presents a number of results and it is unclear why these are f interest or how these impact the detection and/or the prediction process. Perhaps these should be discussed in the discussion section.

7. On page 14, “Although the use of pre-trained W2V word embeddings did not significantly improve the accuracy in the phenotyping task or exceed the accuracy using randomly initialized word embeddings in the predictive task, we have shown in previous unpublished work that W2V can result in faster training requiring fewer epochs during the training of classifiers. This effect was not evaluated in this study.” perhaps should be removed as it relies on unpublished work.

Reviewer #3: This research paper attempted to validate a previous deep learning approach they had developed to detect and predict intentional self-harm using clinical note text. A secondary aim was to assess a set of ICD codes as a silver standard for training models on the automated detection of intentional self-harm and to predict future encounters of ISH.

Some questions below:

1. I’m getting a little lost in some circular logic in terms of clinical validity and utility that mostly stem from use of the ICD codes.

a. Minor comment: the term “well-defined set” is vague. Is it the full list from Hedegaard, 2018?

b. As stated in that same paper and then found extensively in the suicide NLP literature, ICD codes are very incomplete in detecting suicidality. Perhaps self-harm behavior and suicide attempt is a bit better than suicide ideation, but it is still known to have very low sensitivity for identifying cases, with less than half (and sometimes even much, much less than that) of cases detected by ICD codes. The authors’ manual validation of cases and controls as identified by ICD codes is in stark contrast to any literature known by this reviewer. That begs the question of why. Is it the underlying cohort of patients/corpus of notes? The paper does not specify the details of the underlying corpus of 1,538 and 3,012 cases and controls. Is this from the entire EHR in the specified date range? Perhaps the manual review for the controls was affected by ISH being a rare event so not enough notes were seen? Perhaps the control group should be enriched to get a better sense of the false-negative rate of ICD code detection. Another thing that made this reviewer question the validity of the cases and controls was the supplemental material that indicated that many of the ISH encounter notes were from things like anesthesia pre-op OR nursing, and procedure notes. That seems quite odd that discussion and/or coding of self-harm would come up in this setting. During a surgical procedure? Or while the anesthesiologist is seeing the patient for 5 minutes to brief them? I would be curious about the strings that are being reviewed by the manual reviewers from these notes. Are the strings extracted out of order from the original note? Perhaps there is something that is causing a lot of false-positive mentions here… If one of the objectives of the paper is to validate the ICD list as a silver standard (which again goes against the literature), there needs to be more discussion here to establish credibility.

c. If the determination is that ICD codes have precision of 0.8 and recall of 0.99 (which again seem suspect), those accuracy scores are much greater than the deep learning approaches. What then is the rationale for using the deep learning approach?

2. Question regarding concurrent ISH: This reviewer is unclear if this means the ISH event occurred leading up to this encounter? That is what the text seems to imply. But it’s impossible to distinguish from it just being the first time that the ISH is known and documented, right? Like if it’s their first time seeing a psychiatrist and the psychiatrist asks about history of suicide attempts even though it happened 10 years ago? Or do you exclude historic mentions somehow?

3. Along the same lines, how were negated mentions accounted for? Many clinicians might document the absence of ISH.

4. The justification sentence: “However, current approaches to suicide risk assessment involve healthcare professionals administering clinical interviews and questionnaires, which can be inefficient, costly, and limited in their ability to predict future ISH and suicide deaths.” This is circular logic since the notes themselves are documentation of the clinical interviews and assessments. We wouldn’t want clinicians to stop doing what they do—assessing patients. And also if the theoretically did that, what would the algorithm use to predict, since it is relying on mentions of “suicid” etc.?

5. This implication in the Clinical Implications section makes a lot of sense and could be emphasized more including in the objectives: “This would be especially useful in cases where ISH risk is documented in prior clinical notes but is not added to a patient’s primary problem list, and therefore not easily viewable by providers who were not the original documenter.”

6. Later on in the same section: “also the observation and adjustment of those risk levels over time” – not sure what that means.

7. In the limitations section: not sure what this means: “This highlights the need to examine this approach in the using data on fatalities resulting from suicide.”

8. This reviewer is not sure how similar/different this approach was to the original 2020 article from which it drew its methods. It would be helpful to know what has changed and what is different. And why the authors chose to continue these now outdated techniques and to not use language models. Not necessarily saying language models are always better, but would be an important discussion.

9. Finally, the 0.61-0.64 F1 scores are not so good… What does that mean about how to interpret this work and future steps?

6. PLOS authors have the option to publish the peer review history of their article (what does this mean? ). If published, this will include your full peer review and any attached files.

**Do you want your identity to be public for this peer review?** For information about this choice, including consent withdrawal, please see our Privacy Policy .

Reviewer #1: No

Reviewer #2: No

Reviewer #3: No

---

## [Author Response · Author response to Decision Letter 1]

14 Feb 2025

We thank the reviewers for their valuable comments. We have revised the manuscript according to the comments and have provided a detailed response to every comment received in the "Response to Reviewers" document. We believe that we have addressed the concerns and that the revised manuscript is much improved.

---

## [Decision Letter · Decision Letter 1]

5 Jun 2025

PONE-D-24-31813R1Automated detection and prediction of suicidal behavior from clinical notes using deep learningPLOS ONE

Dear Dr. Tsalatsanis,

Thank you for submitting your manuscript to PLOS ONE. After careful consideration, we feel that it has merit but does not fully meet PLOS ONE’s publication criteria as it currently stands. Therefore, we invite you to submit a revised version of the manuscript that addresses the points raised during the review process.

We look forward to receiving your revised manuscript.

Kind regards,

Braja Gopal Patra, Ph.D.

Academic Editor

PLOS ONE

Journal Requirements:

Reviewers' comments:

Reviewer's Responses to Questions

**Comments to the Author**

1. If the authors have adequately addressed your comments raised in a previous round of review and you feel that this manuscript is now acceptable for publication, you may indicate that here to bypass the “Comments to the Author” section, enter your conflict of interest statement in the “Confidential to Editor” section, and submit your "Accept" recommendation.

Reviewer #1: (No Response)

Reviewer #2: All comments have been addressed

Reviewer #3: (No Response)

2. Is the manuscript technically sound, and do the data support the conclusions?

Reviewer #1: Partly

Reviewer #2: (No Response)

Reviewer #3: Yes

3. Has the statistical analysis been performed appropriately and rigorously? 

Reviewer #1: Yes

Reviewer #2: (No Response)

Reviewer #3: I Don't Know

4. Have the authors made all data underlying the findings in their manuscript fully available?

Reviewer #1: No

Reviewer #2: (No Response)

Reviewer #3: No

5. Is the manuscript presented in an intelligible fashion and written in standard English?

Reviewer #1: Yes

Reviewer #2: (No Response)

Reviewer #3: Yes

6. Review Comments to the Author

Reviewer #1: Overall re-review: While I appreciate the authors' response to the previous round of reviews and the corresponding revisions to the paper, I still have some concerns about the overall significance of the study in relation to the claims that are being made. A stronger emphasis on process replicability cf model replicability is required, and more discussion of the limitations of the process and particularly the data selection methodology is required. i do agree that it is valuable to explore the application of the process in a new data context, but claims related to the "high AUC" should be moderated given the unnatural data distribution adopted for evaluation.

[Original] REVIEWER COMMENT: Given the predominance of ED notes for Cases cf Controls, why

was a "matched control" not one that had a similar notes profile to a Case, in terms of type of

notes, number of notes, length?

AUTHOR RESPONSE: The reason for not having similar note profiles between cases

and matched controls is because we used the nearest neighbor matching algorithm, which

prioritizes cases based on proximity in selected characteristics rather than aiming to

create an identical profile for all features, such as type, number, and length of notes.

[New] REVIEWER COMMENT: I find that this is not satisfactorily addressed. The differences in data between cases and matched controls would seem to bias the model, i.e. making it easier to detect the cases based on arguably spurious factors. I understand that the characteristics that were selected include "age, gender, race, ethnicity, and number of notes" -- so number of notes is considered, but only as one of 5 factors, of which 3 are categorical demographic features and therefore have less variance (so will dominate the matching). It would have been more interesting to identify controls that are more likely to be confusers, e.g. those with ICDs for other non-ISH psychological presentations, or as previously suggested that align more strongly in terms of the types of note -- noting the substantial differences between cases and controls described in the section "Clinical Note Characteristics". Clearly if the cases and controls have very different types of notes then distinguishing them shouldn't be very difficult for the model.

In general, more error analysis would help to elucidate what the model is actually learning. For example, for the "39 were diagnosed with ICD codes for suicide attempts or ISH when it was not documented in their clinical notes (i.e., False Positives)" and the "2 patients were not assigned ICD codes for suicide attempts or ISH when their clinical notes indicated they should have (i.e., False Negatives)." -- what was the performance of the model on these cases? Did the models also get confused by the FP cases, and did they detect the FN cases?

[New] REVIEWER COMMENT on Model Validation:

The authors have responded to the concern raised by both reviewers about lack of *model* validation (cf process validation) by citing (a) funding limitations, and (b) challenges of data sharing. I do not understand why data sharing is relevant -- you simply need to share the *trained model* from one hospital, and apply it to the other data. This does not require sharing any data itself. You have access to and have already prepared both data sets; it should be straightforward to take a trained model and apply it to the test data from the other hospital. This is far less time-consuming or costly than the manual analysis that you have added.

[New] REVIEWER COMMENT on prior work:

I appreciate the added paragraph on page 4 with prior work; individual cited work should have the relevant citation numbers added there, e.g. "Cusick and colleagues (2021)^35". Note, however, that here and elsewhere (e.g. p3) it is not meaningful to compare AUC for different task framing and different input data; the inherent difficulty of prediction will be very different from that of detection, and the difficulty of detection will vary depending on how much data is available (e.g. a single ED triage note cf. 6 months of clinical notes). The added statement "the predictive model's performance (especially as highlighted with the AUC) is fairly competitive with what has been previously reported in the literature using different approaches with EHR data, including structured data." really isn't meaningful without more explicit clarification of comparibility of the task settings.

[New] REVIEWER COMMENT on index event:

"Notes recorded within 24 hours of the index event" -- does this mean 24 hours before or after the index event? I assume before, given how the data was collected, but this should be clarified.

REVIEWER RESPONSE to Author response "the replication of the prior methodology is significant as this is a core tenant of the scientific method." I humbly submit that in the context of data-driven methods, the authors' argument here about scientific method is not the key concern. What you are measuring in this work is not the inherent validity of the method, which requires some extrinsic truth to hold. You are measuring the stability of the method as applied in two different data contexts. However, if the method is biased, and the data selection methods are biased, two similar results only reflect consistency in the bias, not evidence that the method is "valid". The similar performance of the method in these two distinct data contexts may be a result of the process itself, not the inherent correctness of the method.

Neither the original work nor this replication are applied prospectively under the condition of real-world data distribution. Rather, both are evaluated only with highly controlled (and similarly biased) selection of cases and controls. The authors have stated as justification for doing this "However, at a single institution, we do not have sufficient data to develop and test the models in a real-world environment." -- yet, they have down-selected controls substantially (although arguably even at 5:1 it would not reflect the true distribution) so they could use this to test in a less biased data set.

Note also you do have to focus on replication rather than reproduction, given that you are not reproducing the methodology on the same dataset. Please review your terminology. As the other reviewer has highlighted as well, given that your focus is on replicating a *process* rather than the study itself this should be clearly identified. This is distinct even from the typical discussions of reproducibility vs replicability of models or their findings (see e.g. https://doi.org/10.1001/jama.2019.20866 )

Reviewer #2: (No Response)

Reviewer #3: Authors have greatly improved the manuscript and this is rigorous and sensical work!

A couple comments/questions (in order of importance):

1. Regarding the ICD Validation through Manual Review on page 12, and the implications.

Previous work has shown that ICD codes are very poor at identifying psychiatric conditions, including suicidality, so it was surprising to see only 2 false negatives. How do the authors explain this unusual finding and what does it mean for the generalizability of the results? If ICD codes have such a high recall in particular, why even the need to develop alternative methods using unstructured data?

2. Page 3, lat paragraph. When discussing the current approaches to suicide risk assessment... Healthcare professionals administering clinical interviews, and then documented in the notes and structured data (and clinical decision making such as to admit or send to ED, etc.) is exactly what the EHR data is from. The existence of EHR data means that encounter happened, and happened well. For example, page 4, "abundance of valuable and relevant information within the EHRs" -- that is put there by clinicians after a clinical assessment. From reading the discussion, seems that not having to *repeat* the clinical interview/questionnaires in different treatment settings (or highlight in more in the chart) after it's already been done... that's the utility! That could be explicit in the intro & discussion.

No other comments!

7. PLOS authors have the option to publish the peer review history of their article (what does this mean? ). If published, this will include your full peer review and any attached files.

**Do you want your identity to be public for this peer review?** For information about this choice, including consent withdrawal, please see our Privacy Policy .

Reviewer #1: No

Reviewer #2: No

Reviewer #3: No

---

## [Author Response · Author response to Decision Letter 2]

18 Jul 2025

Please see the attatched document

---

## [Editor Report · Decision Letter 2]

17 Aug 2025

Automated detection and prediction of suicidal behavior from clinical notes using deep learning

PONE-D-24-31813R2

Dear Dr. Tsalatsanis,

We’re pleased to inform you that your manuscript has been judged scientifically suitable for publication and will be formally accepted for publication once it meets all outstanding technical requirements.

Kind regards,

Braja Gopal Patra, Ph.D.

Academic Editor

PLOS ONE

Additional Editor Comments:

It would be great if you could make the code available on GitHub.

---

## [Editor Report · Acceptance letter]

PONE-D-24-31813R2

PLOS ONE

Dear Dr. Tsalatsanis,

I'm pleased to inform you that your manuscript has been deemed suitable for publication in PLOS ONE. Congratulations! Your manuscript is now being handed over to our production team.

Kind regards,

on behalf of

Dr. Braja Gopal Patra

Academic Editor

PLOS ONE